# Serum Sclerostin But Not DKK-1 Correlated with Central Arterial Stiffness in End Stage Renal Disease Patients

**DOI:** 10.3390/ijerph17041230

**Published:** 2020-02-14

**Authors:** Chun-Feng Wu, Jia-Sian Hou, Chih-Hsien Wang, Yu-Li Lin, Yu-Hsien Lai, Chiu-Huang Kuo, Hung-Hsiang Liou, Jen-Pi Tsai, Bang-Gee Hsu

**Affiliations:** 1Division of Metabolism and Endocrinology, Department of Internal Medicine, Dalin Tzu Chi Hospital, Buddhist Tzu Chi Medical Foundation, Chiayi 62247, Taiwan; Paul210116@hotmail.com; 2Division of Nephrology, Buddhist Tzu-Chi General Hospital, Hualien 97004, Taiwan; simianlkive@gmail.com (J.-S.H.); wangch33@gmail.com (C.-H.W.); nomo8931126@gmail.com (Y.-L.L.); hsienhsien@gmail.com (Y.-H.L.); hermit.kuo@gmail.com (C.-H.K.); 3School of Medicine, Tzu-Chi University, Hualien 97004, Taiwan; 4Division of Nephrology, Department of Medicine, Hsin-Jen Hospital, New Taipei City 24243, Taiwan; hh258527@ms23.hinet.net; 5Division of Nephrology, Department of Internal Medicine, Dalin Tzu Chi Hospital, Buddhist Tzu Chi Medical Foundation, Chiayi 62247, Taiwan

**Keywords:** arterial stiffness, carotid–femoral pulse wave velocity, end-stage renal disease, Dickkopf-1, hemodialysis, peritoneal dialysis, aclerostin

## Abstract

Sclerostin and dickkopf-1 (DKK1) played a role in the development of cardiovascular diseases and arterial stiffness in chronic kidney disease (CKD) patients but with controversial results of patients in end-stage renal disease (ESRD) including hemodialysis (HD) and peritoneal dialysis (PD). This study aimed to examine the association between the mode of dialysis or the values of sclerostin or DKK1 and carotid-femoral pulse wave velocity (cfPWV) in ESRD patients. There were 122 HD and 72 PD patients enrolled in this study. By a validated tonometry system, cfPWV was measured and then segregated patients into values of >10 m/s as the high central arterial stiffness (AS) group and values ≤ 10 m/s as the control group. Serum levels of sclerostin and DKK1 were measured using a commercial enzyme-linked immunosorbent assay kit. Possible risk factors for the development of AS were analyzed by logistic regression analysis. There were 21 (29.2%) of PD and 53 (43.4%) of HD in the high AS group. Compared to patients in the control group, those in the high AS group were older, had more comorbidities, had higher systolic blood pressure, and had higher serum levels of fasting glucose, C-reactive protein, and sclerostin. Levels of sclerostin (adjusted OR 1.012, 95% CI. 1.006–1.017, *p* = 0.0001) was found to be an independent predictor of high AS in ESRD patients by multivariate logistic regression analysis. Furthermore, receiver operating characteristic curve analysis showed the optimal cut-off values of sclerostin for predicting AS was 208.64 pmol/L (Area under the curve 0.673, 95% CI: 0.603–0.739, *p* < 0.001). This study showed that serum levels of sclerostin, but not DKK1 or mode of dialysis, to be a predictor for high central AS in ESRD patients.

## 1. Introduction

Chronic kidney disease (CKD) patients, including those receiving peritoneal dialysis (PD) or maintenance hemodialysis (HD), were well known to have cardiovascular disease (CVD) as their main cause of mortality [1]. Taiwan had the highest incidence and prevalence rate of end-stage renal disease (ESRD) based on the report from the United States Renal Data System. The growing rates were 2.66 and 3.46 times of incidence (from 126 to 331/million populations) and prevalence (from 382 to 1322/million populations) from 1990 to 2001, according to Hwang’s report [2,3], which would further cause a heavy burden of managing these comorbidities of patients. Association with CVD were traditional risk factors including age, hypertension (HTN), diabetes mellitus (DM), non-traditional factors such as inflammation, abnormal bone and mineral metabolism, and vascular calcification [4]. Stiffening of aorta and central large arteries, known as arterial stiffness (AS), had been demonstrated to be an independent risk factor for CVD events by several longitudinal epidemiological studies [5]. There is evidence showing that AS indicated by pulse wave velocity (PWV), which has been considered as a non-invasive gold standard method to access vascular function and as the surrogate marker of arterial wall function and structure. This is a strong predictor of CV events and mortality in ESRD patients independent of classical cardiovascular (CV) risk factors [5,6]. Increasing evidence indicated numerous proteins such as alkaline phosphatase, fetuin-A, parathyroid hormone, C-reactive protein (CRP), abnormal calcium and phosphate homeostasis, osteoprotegerin, and sclerostin were associated with promoting or inhibiting the process of vascular calcification [7,8,9].

Two recently intensely studied factors known as sclerostin and Dickkopf-related protein 1 (DKK1) could modulate the Wnt/β-catenin signaling by binding to LRP5/6 co-receptors to affect bone formation and vascular calcification [8,10,11,12,13,14]. Evidence had shown that decreased expression of sclerostin or DKK1 could lead to an increase in bone formation and even bone overgrowth to cause diseases [10]. Additionally, sclerostin was shown to be associated with vascular smooth muscle calcification in vitro [11], and was positively associated with vascular calcification in patients with DM [12] or CKD [8,9,10,11,12,13,14]. Nevertheless, there were opposite studies showing an inverse relationship between sclerostin and vascular calcification or CV events in CKD and HD patients [15,16,17]. Regarding DKK1, the relationship with vascular calcification was even more controversial with some studies showing no association in HD patients [8,15] or an inverse relationship in elderly men [18]. Since increased PWV had been shown to predict CV morbidity and mortality, the roles of sclerostin and DKK1 in inducing vascular calcification remain inconclusive and the relationship between serum sclerostin and DKK1 with cfPWV in ESRD patients still remained controversial, the aim of this study was to examine the association of sclerostin and DKK1 with central arterial stiffness and to evaluate whether other risk factors, including a different modality of dialysis, would influence AS in ESRD patients.

## 2. Results

Figure 1 showed the methods of sorting patients into control and high AS groups. Demographic, biochemical, and clinical characteristics of all patients, HD and PD patients were individually shown in Table 1, Table 2, and Table 3. Among these patients, there were 101 (52.1%) females and the average age was 61 (IQR 52–71) years old with a median dialysis duration of 48.5 (IQR 22–96) months. A total of 64 patients (33%) had DM, 25 (12.9%) had HTN, and 57 (29.4%) had both. The median values of intact parathyroid hormone (iPTH) and CRP were 230.15 (IQR 89.91–486.30) pg/mL and 0.32 (IQR 0.09–0.90) mg/dL, respectively. The values of sclerostin and DKK1 were 143.50 (IQR 97.21–191.7) pg/mL and 12.08 (IQR 7.15–19.93) pg/mL, respectively (Table 1). The dialysis clearance of HD patients shown as Kt/V and urea reduction ratio were 1.34 ± 0.17 and 0.73 ± 0.04, respectively (Table 2). The dialysis clearance of PD patients shown as weekly Kt/V and total clearance of creatine were 2.09 ± 0.43 and 59.66 ± 24.23, respectively (Table 3).

Compared to the control group, there were 74 patients (38.1%) in the high central AS group. The patients in this group were found to be older, and had higher fasting glucose, higher BP, and higher serum values of CRP and sclerostin regardless of HD or PD among all patients. There were no statistically significant differences in gender, dialysis duration, dialysis adequacy, serum values of iPTH, DKK1, or medications use between the two groups (Table 1). Figure 2 showed median levels of sclerostin in all patients, HD patients, and PD patients, respectively.

After analyzing by univariate and multivariate stepwise logistic regression analyses adjusted by the factors with age, glucose, comorbidity, mode of dialysis, SBP, sclerostin, CRP, gender, and dialysis duration (Table 4), it showed that sclerostin (aOR 1.012, 95% CI 1.006–1.017, *p* = 0.0001), age (aOR = 1.064, 95% CI: 1.032–1.098, *p* = 0.0001), CRP (aOR 2.109, 95% CI 1.196–3.717, *p* = 0.010), SBP (aOR 1.017, 95% CI 1.003–1.032, *p* = 0.018), and glucose (aOR = 1.008, 95% CI: 1.002–1.015, *p* = 0.016) were independent predictors of high AS.

By using the receiver operating characteristic (ROC) curve to predict high AS, it showed that the best cut-off serum value of sclerostin, age, SBP, CRP, and glucose were 208.64 pmol/L, 59 years old, 132 mmHg, 0.2 mg/dL, and 142 mg/dL with area under the curve (AUC) 0.673 (95% CI 0.603–0.739, *p* < 0.0001), 0.672 (95% CI 0.601–0.737, *p* < 0.0001), 0.633 (95% CI 0.561–0.701, *p* = 0.001), 0.664 (95% CI 0.593–0.730, *p* < 0.0001), and 0.641 (95% CI 0.569–0.708, *p* = 0.0006), respectively (Table 5 and Figure 3).

## 3. Discussion

The major findings of the present study were that, regardless of the mode of dialysis, a high serum sclerostin value but not DKK1 was associated with the development of AS and could be a predictor for high central AS in ESRD patients.

Arterial stiffness caused by vascular calcification was found to occur frequently in CKD patients and the incidence increased as renal function progressively declined and was considered a risk factor for CV events for ESRD patients [7,19]. Similar to the analysis showing that AS measured by PWV could be a strong predictor for future CV events in HD patients, Sipahioglu et al. reported that aortic AS could independently predict fatal and non-fatal CV events in PD patients [20]. Szeto et al. also reported that a high baseline value of the cfPWV measurement was associated with worse survival and the longitudinal changes of cfPWV had a significantly positive correlation with SBP in Chinese PD patients [21]. Besides those deranged mineral metabolisms that cause elastin fragmentation and especially medial layer calcification in CKD, evidence had shown that aging might additionally contribute to structural and functional changes of vessels [5,16,22]. In a study conducted in HD patients, London et al. found that there was age-associated increases in aortic PWV, increases in aortic bifurcation diameter, a decrease in an aortic taper, a lower brachial/aortic stiffness gradient, and a decline that is more significant with age, which were indicative of more marked age-related AS in ESRD patients [22]. Chi et al. similarly found that old age was a significant risk factor for central AS in Taiwan PD patients [23]. Traditional CV risk factors, including DM, hyperlipidemia, and elevated BMI had been related to AS [18], and a cross-sectional study showing that, after adjusting for confounders, DM was independently associated with increased incidence of central AS [24]. Schram et al. showed that patients with impaired glucose metabolism also had higher risk of developing central AS [24]. Moreover, it had been known that vascular calcification is a process of gradual osteogenesis initiated by inflammatory factors in vessels and it was found that higher serum CRP was associated with the presence of aortic calcification in CKD patients [17] as well as in HD and PD patients in addition to old age, hypoalbuminemia, and DM [16,25]. Taken together, we found that ESRD patients who were older, had higher glucose, and higher CRP levels were associated with a higher degree of cfPWV, which indicated that these variables might be possible factors modulating the process of AS.

After being discovered in 2001 as a 22-kD glycoprotein encoded by the SOST gene, which functioned as an osteocyte-derived bone morphogenetic protein and as an antagonist through the Wnt signaling pathway to inhibit differentiation, proliferation, and promote osteoblast apoptosis [26,27], the roles of sclerostin in vascular calcification and clinical outcomes was studied extensively in kidney disease patients [13,14,28,29]. Evidence showed that, compared to those with normal renal function, serum levels of sclerostin started to increase from CKD stage III and progressively increased to two to four times higher as renal function progressed to ESRD needing dialysis [28,29]. Bonani et al. found that the serum concentration of sclerostin rapid lowered and paralleled the improvements of renal function after receiving renal transplantation and proposed that decreased renal clearance might be responsible for the accumulation of sclerostin in late stage of CKD [30]. Additionally, there was evidence showing no correlation between vascular SOST mRNA expression and circulating sclerostin levels, which indicated less contribution of vascular-derived sclerostin [14]. By binding to LRP5/6 or forming a tertiary complex with LRP5/6, sclerostin and DKK1, respectively, exerted their role in modulating the cross-talk of CKD-related bone disease and vasculatures with the results of vascular smooth muscle cells trans-differentiation and vascular calcification [7,11,15,31]. Thambiah et al. and Kuo et al. conducted studies in patients of advanced renal disease and PD and found a positive correlation between serum levels of sclerostin and bone mineral density (BMD) [31,32]. By intervention with an anti-resorptive agent, Miyaoka et al. found denosumab had reno-protective effects with the reverse of the age-related decrease of eGFR by inhibiting bone resorption and suppressing seral phosphate load in non-CKD patients [33], which indicated a possible role of manipulating sclerostin toward renal or vascular protection. From the reports European Uremic Toxin Work Group, sclerostin was found to be positively associated with a decline of eGFR, phosphate, HDL, interleukin 6, CRP, free indoxyl sulfate, p-cresyl sulfate, and cfPWV, which indicated its link with inflammation, vascular damages, and bone diseases of CKD patients [34]. Arterial stiffness caused by vascular calcification was found to occur frequently in CKD patients and the incidence increased as renal function progressively declined [4,7]. Furthermore, vascular calcification was considered as an imbalance between the inhibitors and promoters of osteogenesis initiated in vessels by uremic factors of CKD patients [7]. Among these factors, evidence had shown that sclerostin played an important role and increased expression of sclerostin was associated with vascular calcification [13,14,28,29]. Furthermore, Lv et al. and Morena et al. stratified CKD patients according to median or tertiles plasma sclerostin level and showed that those with values of sclerostin of more than 59.1 pmol/L or 0.748 ng/ml were more likely to have a decline of renal function along with vascular calcification and coronary artery calcification, respectively [8,13]. Qureshi et al. reported that serum sclerostin was related to vascular calcification with a significant predictive value (AUC 0.68) of scoring coronary and epigastric artery calcification in ESRD patients [14]. Moreover, there was a significant positive correlation between serum levels of sclerostin and peripheral AS measured by brachial-ankle PWV in renal transplantation recipients in our previous study [35]. In agreement with these studies, we found that serum levels of sclerostin positively correlated with cfPWV and was an independent predictor for development of AS in ESRD patients with an optimal cutoff value of 208.64 pmol/L, regardless of the mode of dialysis. Taken together, we considered that sclerostin might play a role in modulating the development of vascular calcification in impaired renal function patients, including those receiving PD and HD.

DKK1 was secreted by osteoblast and osteocytes as a 26-kD small glycoprotein and functioned as an antagonist of the Wnt/-catenin signaling pathway to modulate vascular ossification [27,36]. Similar to sclerostin, DKK1 was thought to have a role to regulate vascular calcification, but studies with regard to the relationship between DKK1 and vascular calcification remained inconclusive [8,15,18]. By assessing the severity of the lateral spine, Szulc et al. reported that, as the abdominal aorta calcification scores were more severe, the serum levels of DKK1 were lower in men older than 60 years old [18]. Furthermore, there was no association between levels of DKK1 and coronary artery calcification by scoring the coronary artery calcification with a multi-detector computed tomography in CKD patients (median estimated glomerular filtration rate 35.1 ml/min/1.73m^2^) [8]. Moreover, serum levels of DKK1 were also shown to have no association with future CV events or aortic calcification scored by plain film in the maintenance of HD patients [15]. In our previous study conducted in renal transplantation recipients, there was no association between serum levels of DKK1 and peripheral AS indicated by brachial-ankle PWV [35]. Similarly, we found that there was no association between serum levels of DKK1 and cfPWV or central AS in this study. Like these studies, there was no role to predict high AS in our ESRD patients.

The limitation of this study was that it was cross-sectional with a limited number of ESRD patients. Therefore, the relationship of serum sclerostin with AS and the role of sclerostin to predict high AS of ESRD patients should be confirmed by further longitudinal studies before a cause-effect can be established.

## 4. Conclusions

In this study, we found that serum sclerostin was greater than 208.64 pmol/L and, together with old age, high values of CRP, glucose, and SBP were possible predictors for the development of AS in ESRD patients regardless of the modality of dialysis. These findings indicated that sclerostin but not DKK1 might modulate the process of vascular calcification to cause high AS in advanced renal disease patients, but the mechanism remained to be studied.

## 5. Materials and Methods 

### 5.1. Patients

In this study, calculation of sample size was based on studies of Lv et al. and Qureshi et al. [13,14]. After input of α (type I error) 0.05, β (type II error) 0.9, and correlation coefficient (β, 0.36) [13] or AUC (0.68) [14], the needed sample size for study was 77 [13] or 109 (needed number of positive or negative cases required were 45 and 64) [14], respectively. Therefore, we enrolled all ESRD patients (120 and 74 patients of control and high AS groups) if they agree to participate this study and did not have an acute infection, malignancy, acute myocardial infarction, pulmonary edema, or heart failure at the time of blood sampling.

This was a cross-sectional study, which enrolled 72 PD and 122 HD patients from Hualien and Dalin Tzu Chi Hospital from March 2015 to October 2016. All patients underwent regular PD for more than 3 months. Among these patients, 52 received continuous ambulatory PD (CAPD, Dianeal, Baxter Health Care, Taiwan), with three to five dialysate exchanges per day while 20 other patients underwent four to five dialysate exchanges each night with an automated device (automated PD, APD). HD patients who were older than 20 years of age, having received standard 4-hour dialysis three times per week for at least three months using standard bicarbonate dialysate were enrolled. All patients received disposable high flux polysulfone artificial kidney (FX class dialyzer, Fresenius Medical Care, Bad Homburg, Germany).

Blood pressure (BP) of all patients was measured by trained staff in the morning using standard mercury sphygmomanometers with appropriate cuff sizes after sitting for at least 10 minutes. Systolic BP (SBP) and diastolic BP (DBP) were taken three times at 5-minute intervals and were averaged for analysis. Patients who were diagnosed to have HTN were defined as SBP≥140 mmHg, and/or DBP≥90 mmHg, or have received any anti-hypertensive medication in the past two weeks. Patients were regarded as DM if the fasting plasma glucose was either 126 mg/dL or more or using oral hypoglycemic medications or insulin. Patients who agreed for study received anthropometric and cfPWV examinations and had their blood samples collected, centrifuged, stored, and investigated in the same day. Samples for measuring sclerostin and DKK1 were collected and stored at the same time and then examined by enzyme-linked immunosorbent assay altogether thereafter.

All participants provided their informed consents before participating in this study. Approval from the Protection of Human Subjects Institutional Review Board of Tzu-Chi University and Hospital was obtained. 

### 5.2. Anthropometric Analysis

All anthropometric factors were measured three times after overnight fasting and without dialysate in the abdominal cavity of PD patients. Bodyweight (BW) and height (BH) were measured in light clothing and without shoes to the nearest 0.5 kg and 0.5 cm. Body mass index (BMI) was calculated as BW (kg)/BH^2^ (m^2^). Body fat mass measured by bioimpedance was performed by a standard tetrapolar whole body (hand-foot) technique, using a single-frequency (50-kHz) analyzer (Biodynamic-450, Biodynamics Corporation, Seattle, WA, USA) and analysed by a specific formula supplied by the manufacturer [9,35,37]. All patients were measured by the same operator.

### 5.3. Biochemical Investigations

Blood samples (approximately 5 mL) were collected and were immediately centrifuged at 3000× *g* for 10 minutes in the morning before exchanging PD dialysate or receiving HD. Serum samples were stored at 4° C and used for biochemical analyses within 1 h of collection. The fractional clearance index for urea (Kt/V) and urea reduction ratio of HD patients were measured using a formal, single-compartment dialysis urea kinetic model. The weekly fractional clearance index for urea (weekly Kt/V), total clearance of creatinine, and peritoneal clearance of creatinine of PD patients were provided from medical records. Serum levels of blood urea nitrogen (BUN), creatinine, fasting glucose, albumin, total cholesterol (TCH), triglyceride (TG), total calcium, phosphorus, and CRP were measured using an autoanalyzer (Siemens Advia 1800, Siemens Healthcare GmbH, Henkestr, Germany). Serum values of sclerostin and DKK1 levels (Biomedica immunoassays, Vienna, Austria) and intact parathyroid hormone (iPTH) (Diagnostic Systems Laboratories, Webster, TX, USA) were quantified by using commercially enzyme-linked immunosorbent assays [35]. The intra-assay and inter-assay coefficients of variation in the measurement for the sclerostin level was 4.5% and 6.1% and DKK1 levels were 2.2% and 2.9%, respectively. The number of human sclerostin and DKK1 ELISA kit catalog was BI-20492 and BI-20413, respectively.

### 5.4. Carotid-Femoral Pulse Wave Velocity Measurements

Carotid-femoral PWV (cfPWV) was measured by an applanation tonometry (SphygmoCor system, AtCor Medical, Australia) as shown in previous studies [9,35,37]. These measurements were performed in all patients in the supine position after a minimum of 10 minutes rest in a quiet and temperature-controlled room after having breakfast in the morning. Records were made simultaneously with an electrocardiography (ECG) signal, which provided an *R*-timing reference. Pulse wave recordings were performed consecutively at two superficial artery sites (carotid-femoral segment). Integral software was used to process each set of pulse wave and ECG data to calculate the mean time difference between the *R*-wave and the pulse wave on a beat-to-beat basis, with an average of 10 consecutive cardiac cycles. The cfPWV was calculated using the distance and mean time difference between the two recorded points. Quality indices, included in the software, were set to ensure uniformity of data. In this study, cfPWV values of >10 m/s were defined as a high AS group and those ≦10 m/s were defined as the control group, according to the European Society of Hypertension (ESH) and the European Society of Cardiology (ESC) guidelines [38].

### 5.5. Statistical Analysis

Continuous variables were analysed for normal distribution by using the Kolmogorov–Smirnov test. Continuous variables were expressed as the mean ± standard deviation (SD) or medians and interquartile ranges (IQR) and comparisons between patients were analyzed by using the Student’s independent t-test (two-tailed) or the Mann-Whitney U test, accordingly. Categorical variables were expressed as the number of patients and were analyzed by the chi-squared test. Mode of dialysis was further analyzed by a chi-squared test with a continuity correction. Risk factors for the development of AS were analyzed by using univariate and multivariate stepwise logistic regression analysis. Variables showed significance in univariate logistic regression analysis and those baseline characteristics with *p* < 0.1 are shown in Table 1 (age, glucose, comorbidity, mode of dialysis, SBP, sclerostin, CRP, gender, and duration of dialysis) were used in multivariate logistic regression analysis. A receiver operating characteristic (ROC) curve was used to calculate the area under the curve (AUC) to identify the cut-off value of variables to predict high AS in ESRD patients. All data were analyzed by using SPSS for Windows (version 19.0; SPSS Inc., Chicago, IL, USA) except the sample size calculation and the ROC curve, which were analyzed by MedCalc^®^ (version 12.7.2.0). Additionally, *p* values < 0.05 were considered statistically significant.

## Figures and Tables

**Figure 1 ijerph-17-01230-f001:**
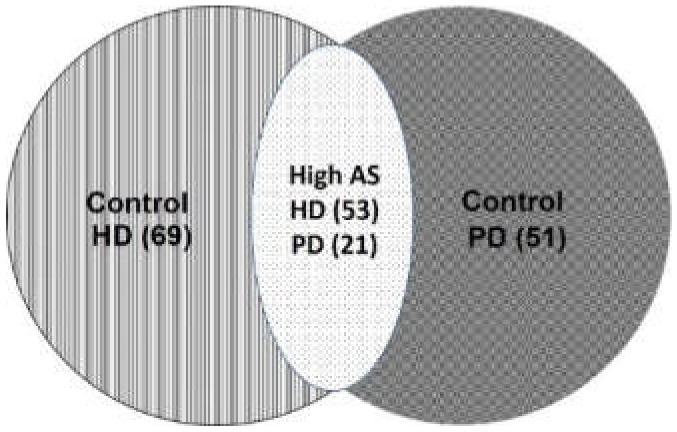
Venn diagram of dialysis patients.

**Figure 2 ijerph-17-01230-f002:**
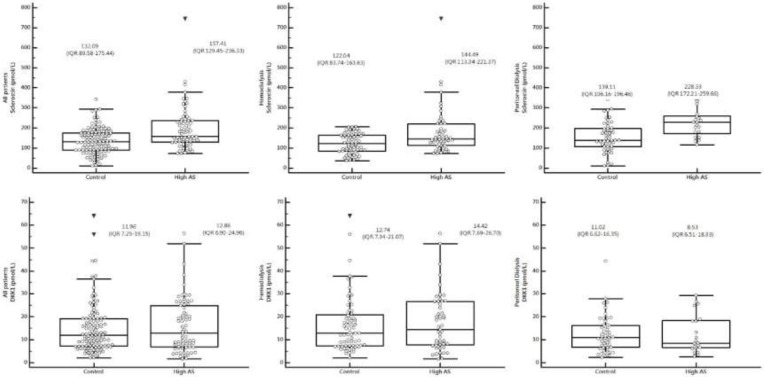
Values of sclerostin and DKK1 shown as box and whisker plots of all patients (**left panel**), HD (**middle panel**), and PD (**right panel**).

**Figure 3 ijerph-17-01230-f003:**
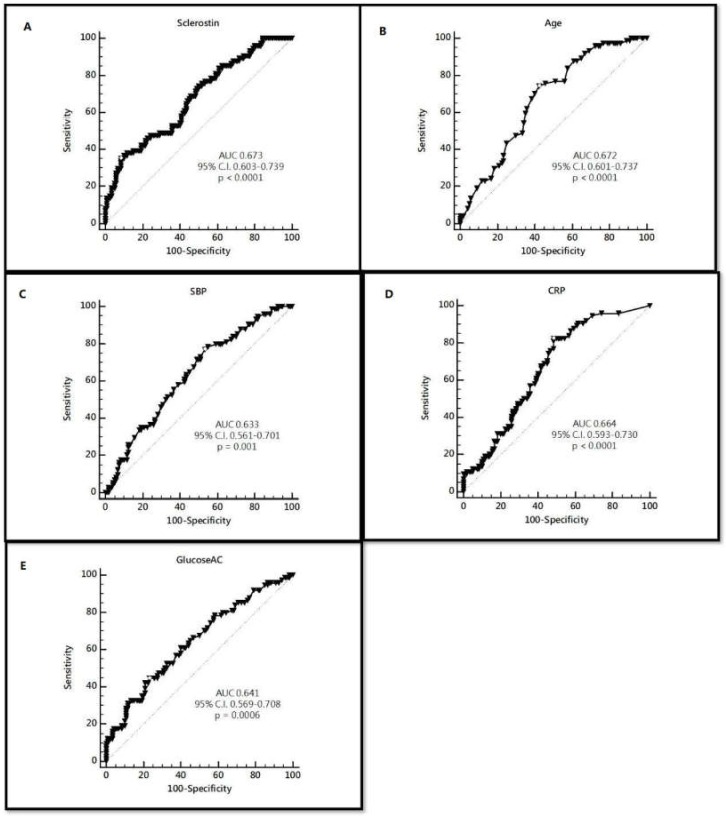
ROC curve and cut-off level of (**A**) sclerostin, (**B**) Age, (**C**) SBP, (**D**) CRP, and (**E**) fasting glucose to predict high arterial stiffness.

**Table 1 ijerph-17-01230-t001:** Clinical variables of the dialysis patients in high AS or control groups.

Characteristics	All (*n* = 194)	Control (*n* = 120)	High AS (*n* = 74)	*p*
Age (years)	61 (52–71)	58 (47.5–67.5)	65 (59 –73)	0.0001
Female, *n* (%)	101 (52.1)	69 (57.5)	32 (43.2)	0.075
Dialysis duration (mo)	48.5 (22–96)	42.5 (18.5–96.5)	57 (28–90)	0.077
BMI (Kg/m^2^)	24.91 ± 4.82	24.58 ± 4.83	25.45 ± 4.80	0.226
SBP (mmHg)	142.29 ± 24.94	138.09 ± 25.54	149.11 ± 22.46	0.003 *
DBP (mmHg)	79.66 ± 15.74	78.81 ± 14.84	81.05 ± 17.12	0.336
cfPWV (m/s)	9.0 (7.5–11.7)	7.7 (7.0–8.9)	12.3 (11.4–14.3)	<0.001 *
BUN (mg/dL)	60 (50–70)	60.0 (49.5–67.5)	61.0 (50.0–71.0)	0.564
Creatinine (mg/dL)	9.91 ± 2.67	9.90 ± 2.83	9.95 ± 2.43	0.901
Calcium (mg/dL)	9.02 ± 0.76	8.96 ± 0.71	9.11± 0.84	0.197
IP (mg/dL)	4.92 ± 1.34	4.94 ± 1.37	4.90 ± 1.29	0.842
Albumin (mg/dL)	4.06 (3.7–4.2)	4.01 (3.70–4.30)	4.10 (3.70–4.20)	0.644
TCH (mg/dL)	154.73 ± 39.13	157.55 ± 40.58	150.15 ± 36.47	0.202
TG (mg/dL)	125 (87–197)	114.0 (87.5–199.0)	133.0 (87.0–189.0)	0.473
Glucose (mg/dL)	120 (100–149)	114.5 (97.0–142.0)	131.0 (110.0–182.0)	0.001
CRP (mg/dL)	0.32 (0.09–0.90)	0.195 (0.06–0.780)	0.465 (0.25–1.05)	0.0001 *
iPTH (pg/mL)	230.15 (89.91–486.30)	252.15 (123.07–503.86)	167.25 (73.01–434.10)	0.111
Sclerostin (pmol/L)	143.50 (97.21–191.7)	132.09 (89.58–175.44)	157.41 (129.45–236.33)	0.0001 *
DKK-1 (pmol/L)	12.08 (7.15–19.93)	11.96 (7.25–19.15)	12.86 (6.90–24.96)	0.563
Mode, n (%)				<0.001
PD	72 (37.1)	51 (70.8)	21 (29.2)	
HD	122 (62.9)	69 (56.6)	53 (43.4)	
Cormobidity n (%)				0.003
No	48 (24.7)	37 (30.8)	11 (14.9)	
Diabetes mellitus	64 (33.0)	44 (36.7)	20 (27.0)	
Hypertension	25 (12.9)	14 (11.7)	11 (14.9)	
Both	57 (29.4)	25 (20.8)	32 (43.2)	
ARB, *n* (%)	66 (34.0)	43 (35.8)	27 (36.5)	0.951
β-blocker, *n* (%)	63 (32.5)	38 (31.7)	25 (33.8)	0.882
CCB, *n* (%)	82 (42.3)	55 (45.8)	27 (36.5)	0.258
Statin, *n* (%)	40 (20.6)	24 (20.0)	16 (21.6)	0.930

BMI, body mass index. cfPWV, carotid-femoral pulse wave velocity. TCH, total cholesterol. TG, triglyceride. HD, hemodialysis. PD, peritoneal dialysis. SBP, systolic blood pressure. DBP, diastolic blood pressure. BUN, blood urea nitrogen. CRP, C-reactive protein. iPTH, intact parathyroid hormone. IP, inorganic phosphate. DKK-1, dickkopf-1. Kt/V, fractional clearance index for urea. ARB, angiotensin receptor blocker. CCB, calcium channel blocker. Continuous variables are shown as mean ± standard deviation or median and interquartile range after analysis by Student’s t-test or Mann-Whitney U test according to the analysis for normal distribution. Categorical variables are presented as number (%) and analyzed by a chi-square test. Mode of dialysis was analyzed by chi-squared with a continuity correction. * *p* < 0.05 was statistically significant.

**Table 2 ijerph-17-01230-t002:** Clinical characteristics of the HD patients in high AS and control groups.

Characteristics	All Patients (*n* = 122)	Control Group (*n* = 69)	High AS Group (*n* = 53)	*p*
Age (years)	63.30 ± 12.14	60.58 ± 13.00	66.83 ± 9.98	0.004 *
Female, *n* (%)	60 (49.2)	37 (53.6)	23 (43.4)	0.263
HD duration (mo)	57.00 (25.53–119.34)	58.20 (21.84–131.94)	56.88 (26.70–104.82)	0.857
BMI (Kg/m^2^)	24.92 ± 5.06	24.63 ± 5.28	25.29 ± 4.78	0.479
DM, *n* (%)	52 (42.6)	19 (27.5)	33 (62.3)	<0.001 *
HTN, *n* (%)	59 (48.4)	27 (39.1)	32 (60.4)	0.020 *
SBP (mmHg)	142.47 ± 25.61	137.67 ± 26.59	148.72 ± 23.05	0.018 *
DBP (mmHg)	76.74 ± 16.40	76.07 ± 15.63	77.60 ± 17.46	0.611
cfPWV (m/s)	10.07 ± 2.98	7.88 ± 1.17	12.92 ± 2.06	<0.001 *
BUN (mg/dL)	61.06 ± 15.61	60.77 ± 14.94	61.43 ± 16.58	0.816
Creatinine (mg/dL)	9.32 ± 2.07	9.36 ± 2.08	9.28 ± 2.09	0.836
Calcium (mg/dL)	9.00 ± 0.74	8.94 ± 0.71	9.07± 0.79	0.331
IP (mg/dL)	4.76 ± 1.26	4.75 ± 1.25	4.79 ± 1.29	0.862
Albumin (mg/dL)	4.17 ± 0.46	4.18 ± 0.47	4.16 ± 0.45	0.840
TCH (mg/dL)	144.65 ± 35.32	147.45 ± 39.38	141.00 ± 29.18	0.320
TG (mg/dL)	113.00 (85.50-187.00)	106.00 (85.00–192.50)	127.00 (85.00–184.00)	0.437
Glucose (mg/dL)	130.50 (117.75–169.00)	128.00 (106.50–153.50)	137.00 (114.00–185.50)	0.084
CRP (mg/dL)	0.41 (0.12–0.92)	0.25 (0.08–0.79)	0.59 (0.25–1.05)	0.003 *
iPTH (pg/mL)	204.05 (84.08–416.65)	244.40 (121.90–445.05)	157.60 (58.00–392.15)	0.180
Sclerostin (pmol/L)	133.54 (90.52–175.17)	122.04 (83.74–163.63)	144.49 (113.34–221.37)	0.002 *
DKK-1 (pmol/L)	13.25 (7.40–22.61)	12.74 (7.34–21.07)	14.42 (7.69–26.70)	0.586
Urea reduction rate	0.73 ± 0.04	0.74 ± 0.04	0.73 ± 0.04	0.689
Kt/V (Gotch)	1.34 ± 0.17	1.35 ± 0.17	1.33 ± 0.16	0.658
ARB, *n* (%)	36 (29.5)	18 (26.1)	18 (34.6)	0.344
β-blocker, *n* (%)	38 (31.1)	19 (27.5)	19 (35.8)	0.326
CCB, *n* (%)	47 (38.5)	30 (43.5)	17 (32.1)	0.200
Statin, *n* (%)	20 (16.4)	9 (13.0)	11 (20.8)	0.254

BMI, body mass index. cfPWV, carotid-femoral pulse wave velocity. TCH, total cholesterol. TG, triglyceride. HD, hemodialysis. SBP, systolic blood pressure. DBP, diastolic blood pressure. BUN, blood urea nitrogen. CRP, C-reactive protein. iPTH, intact parathyroid hormone. IP, inorganic phosphate. DKK-1, dickkopf-1. Kt/V, fractional clearance index for urea. ARB, angiotensin receptor blocker. CCB, calcium channel blocker. DM, diabetes mellitus. HTN, hypertension. Continuous variables are shown as mean ± standard deviation or median and interquartile range after analysis by Student’s t-test or Mann-Whitney U test according to analysis for normal distribution. Categorical variables are presented as number (%) and analyzed by a chi-square test. * *p* < 0.05 was statistically significant.

**Table 3 ijerph-17-01230-t003:** Clinical variable of the PD patients with high AS and control groups.

Characteristic	All Participants (*n* = 72)	Control Group (*n* = 51)	High AS Group (*n* = 21)	*p* Value
Age (years)	54.86 ± 16.11	51.61 ± 16.37	62.76 ± 12.59	0.007*
Female, n (%)	41 (56.9)	32 (62.7)	9 (42.9)	0.121
PD vintage (months)	46.07 ± 38.86	39.76 ± 38.66	61.38 ± 35.73	0.031 *
BMI (kg/m^2^)	24.91 ± 4.43	24.52 ± 4.19	25.85 ± 4.92	0.249
DM, n (%)	30 (41.7)	20 (39.2)	10 (47.6)	0.511
HTN, n (%)	62 (86.1)	42 (82.4)	20 (95.2)	0.151
CAPD, n (%)	52 (72.2)	36 (70.6)	16 (76.2)	0.630
SBP (mmHg)	142.00 ± 23.94	138.67 ± 24.31	150.10 ± 21.42	0.065
DBP (mmHg)	84.63 ± 13.25	82.51 ± 12.95	89.76 ± 12.87	0.034 *
cfPWV (m/s)	9.08 ± 3.12	7.47 ± 1.69	12.98 ± 2.18	< 0.001*
BUN (mg/dL)	59.01 ± 18.65	58.57 ± 18.91	60.10 ± 18.39	0.755
Creatinine (mg/dL)	10.92 ± 3.24	10.62 ± 3.49	11.63 ± 2.45	0.235
Calcium (mg/dL)	9.04 ± 0.80	8.98 ± 0.73	9.18 ± 0.95	0.339
IP (mg/dL)	5.19 ± 1.43	5.20 ± 1.50	5.18 ± 1.28	0.964
Albumin (mg/dL)	3.74 ± 0.38	3.74 ± 0.42	3.73 ± 0.28	0.913
TCH (mg/dL)	171.81 ± 39.58	171.22 ± 38.46	173.24 ± 43.12	0.845
TG (mg/dL)	147.00 (91.50–212.75)	139.00 (91.00–200.00)	159.00 (108.00–232.50)	0.511
Glucose (mg/dL)	105.00 (94.25–126.75)	101.00 (91.00–120.00)	117.00 (100.00–165.50)	0.011 *
iPTH (pg/mL)	250.05 (121.64–558.70)	259.70 (120.20–585.90)	229.20 (111.71–503.42)	0.706
CRP (mg/dL)	0.26 (0.07–0.80)	0.14 (0.06–0.69)	0.32 (0.24–1.07)	0.015 *
Sclerostin (pmol/L)	173.91 (122.32–229.02)	139.11 (106.16–196.46)	228.33 (172.21–259.66)	<0.001 *
DKK-1 (pmol/L)	10.55 (6.63–17.30)	11.02 (6.62–16.35)	8.53 (6.51–18.33)	0.733
Weekly Kt/V	2.09 ± 0.43	2.15 ± 0.45	1.95 ± 0.33	0.070
Peritoneal Kt/V	1.74 ± 0.45	1.76 ± 0.46	1.71 ± 0.42	0.654
Total CCr (l/week)	59.66 ± 24.23	61.54 ± 25.66	55.09 ± 20.20	0.308
Peritoneal CCr (l/week)	42.33 ± 16.34	42.33 ± 16.88	42.35 ± 15.35	0.995
ARB, n (%)	30 (41.7)	22 (43.1)	8 (38.1)	0.693
β-blocker, n (%)	25 (34.7)	19 (37.3)	6 (28.6)	0.482
CCB, n (%)	35 (48.6)	25 (49.0)	10 (47.6)	0.914
Statin, n (%)	20 (27.8)	15 (29.4)	5 (23.8)	0.630

BMI, body mass index. cfPWV, carotid-femoral pulse wave velocity. TCH, total cholesterol. TG, triglyceride. CAPD, continuous ambulatory peritoneal dialysis. SBP, systolic blood pressure. DBP, diastolic blood pressure. BUN, blood urea nitrogen. CRP, C-reactive protein. iPTH, intact parathyroid hormone. IP, inorganic phosphate. DKK-1, dickkopf-1. Kt/V, fractional clearance index for urea. CCr, creatinine clearance rate. ARB, angiotensin receptor blocker. CCB, calcium channel blocker. DM, diabetes mellitus. HTN, hypertension. Continuous variables are shown as mean ± standard deviation or median and interquartile range after analysis by Student’s t-test or Mann-Whitney U test according to analysis for normal distribution. Categorical variables are presented as number (%) and analyzed by a chi-square test. * *p* < 0.05 was statistically significant.

**Table 4 ijerph-17-01230-t004:** Univariate and multivariate stepwise logistic regression analysis of the factors correlated to arterial stiffness among 194 dialysis patients.

Variables	OR	95% CI	*p*	aOR	95% CI	*p*
Age (years)	1.052	1.025–1.079	0.0001 *	1.064	1.032–1.098	0.0001
Female	0.563	0.314–1.011	0.055			
Dialysis duration	1.002	0.997–1.007	0.448			
BMI	1.038	0.977–1.102	0.226			
SBP (mmHg)	1.019	1.006–1.031	0.003	1.017	1.003–1.032	0.018
DBP (mmHg)	1.009	0.991–1.028	0.335			
BUN (mg/dL)	1.004	0.987–1.022	0.622			
Creatinine (mg/dL)	1.007	0.904–1.122	0.900			
Calcium (mg/dL)	1.286	0.877–1.885	0.198			
IP (mg/dL)	0.978	0.787–1.215	0.841			
Albumin (mg/dL)	1.228	0.669–2.257	0.508			
TCH (mg/dL)	0.995	0.988–1.003	0.201			
TG (mg/dL)	0.9996	0.997–1.002	0.764			
Glucose (mg/dL)	1.010	1.004–1.016	0.001	1.008	1.002–1.015	0.016
iPTH (pg/mL)	0.9995	0.98–1.001	0.322			
CRP (mg/dL)	1.915	1.248–2.937	0.003	2.109	1.196–3.717	0.001
Sclerostin (pmol/L)	1.010	1.005–1.014	<0.0001	1.012	1.006–1.017	0.0001
DKK–1 (pmol/L)	1.011	0.985–1.038	0.409			
Mode n (%)						
PD	1					
HD	1.865	1.002–3.473	0.049			
Comorbidity						
No	1					
DM	1.529	0.650–3.598	0.331			
HTN	2.643	0.936–7.460	0.066			
Both	4.306	1.836–10.099	0.0008			
ARB	1.029	0.563–1880	0.927			
β-blocker	1.101	0.594–2.039	0.760			
CCB	0.679	0.374–1.230	0.201			
Statin	1.103	0.542–2.248	0.786			

BMI, body mass index. Alb, albumin. TCH, total cholesterol. TG, triglyceride. HD, hemodialysis. PD, peritoneal dialysis. SBP, systolic blood pressure. DBP, diastolic blood pressure. BUN, blood urea nitrogen. CRP, C-reactive protein. iPTH, intact parathyroid hormone. IP, inorganic phosphate. DKK-1, dickkopf-1. Kt/V, fractional clearance index for urea. ARB, angiotensin receptor blocker. CCB, calcium channel blocker. DM, diabetes mellitus. HTN, hypertension. Analysis data was done using the multivariable logistic regression analysis (adopted factors: Age, Glucose, Comorbidity, Mode of dialysis, SBP, sclerostin, CRP, gender, dialysis duration).* *p* < 0.05 was statistically significant.

**Table 5 ijerph-17-01230-t005:** The best cut-off values of each variables at predicting central arterial stiffness of dialysis patients by receiver operating characteristic (ROC) curve analysis.

Variables	Criterion	Sensitivity	95% CI	Specificity	95% CI
Sclerostin	208.64	35.14	24.4–47.1	91.67	85.2–95.9
Age	59	74.32	62.8–83.8	57.50	48.1–66.5
SBP	132	77.03	65.8–86.0	46.67	37.5–56.0
CRP	0.2	82.43	71.8–90.3	51.67	42.4–60.9
Glucose	142	44.59	33.0–56.6	76.67	68.1–83.9

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
