# Peer review of "Serum Sclerostin But Not DKK-1 Correlated with Central Arterial Stiffness in End Stage Renal Disease Patients"

_ijerph, 2020, doi:10.3390/ijerph17041230_

Round 1
Reviewer 1 Report
Manuscript submitted by Wu et.al., entitled as, “Serum Sclerostin but Not DKK-1 Correlated with Central Arterial Stiffness in End Stage Renal Disease Patients” is a cross-sectional, single center study of end-stage renal disease patients, including hemodialysis and peritoneal dialysis patients. Authors identified sclerostin as a biomarker of high arterial stiffness. The manuscript is well written and study design is acceptable. However, few details are missing, as outlined below.
In the introduction section, authors should outline reasons for discrepancies in past-studies where association of DKK1 and sclerostin with vascular calcification was established. And, what innovation was involved in this study that will make these controversies resolved. Authors claim their study being first of its kind. However, Sclerostin, not DKK1, had been proposed as a biomarker in the pathogenesis of chronic kidney disease-mineral and bone disorder (https://doi.org/10.1371/journal.pone.0176411). Authors should avoid using extreme words like “first of its kind”. Authors should comment on role of sclerostin on CKD, and disease progression. Characterstics of control patients are not clear. On what basis Control patients were excluded from high AS, HD, and PD patients. Are control patients describe in high AS are exclusive of HD and PD patients? E.g. whether all dialysis patients were first classified as High AS patients, and all other dialysis recipient were treated as control? Please explain. A Venn diagram of different patients could be helpful in identifying different group of patients. Authors did not provide timeline of study? Also, how many days patients were examined, or if it was a single point, single day, type of study. The level of Sclerostin and DKK-1 was measured using ELISA based assay. Their values in patients should be shown as “box and whisker plots, with dots for individual value”. This value-distribution is important, as Sclerostin value in Control patients’ ranges from 89 to 175, and in High AS patients’ from 129 to 236 (Table 1). Did authors tried to group patients based on Sclerostin values (level) and determine what value of sclerostin can indicate patient condition well? Figure 1, legend are not self-explanatory. What does each arrowhead represent on the plot? What is the cut-off level for each parameter? What does sensitivity and 100-Specificity mean? How these plots were generated (software used) and what does they mean? Are they of diagnostic value in future? Why cfPWV was not included, as that is of prognostic value in determination of ESRD. In the discussion section, authors should discuss different origin and function of Sclerostin and DKK1 and their implications in ESRD. Line 143-146, authors compared PWV with “aortic stiffness index beta”. What is the implication of it in the present study is not clear. Similarly, several other points are introduced without discussing their implication with the present study. Authors should refrain from cataloguing all the information in the discussion section; rather it should be used for discussing implications of their findings. Authors are encouraged to restructure the discussion section. The most interesting finding of the study is identification of optimal cut-off values for AS prediction. Authors should discuss this finding in more detail. Manuscript needs to be proof-read well for typos, errors in grammatical usage of English, and sentence structure.
Author Response
In the introduction section, authors should outline reasons for discrepancies in past-studies where association of DKK1 and sclerostin with vascular calcification was established. And, what innovation was involved in this study that will make these controversies resolved. Authors claim their study being first of its kind. However, Sclerostin, not DKK1, had been proposed as a biomarker in the pathogenesis of chronic kidney disease mineral and bone disorder (https://doi.org/10.1371/journal.pone.0176411). Authors should avoid using extreme words like “first of its kind”.
Ans: Thanks for your comments.
In the Introduction section, 2nd Paragraph, we described that from previous studies, the correlation between sclerostin and vascular calcification showed controversial results. Some (reference 12-14) showed positive correlation, while others (reference 15-17) showed negative correlation. For DKK1, the association with vascular calcification was even more controversial (reference 15, 18).
Because the relationship between sclerostin and DKK1 and arterial stiffness remained inconclusive and controversial, we performed this study to examine the relationship and additionally we enrolled both HD and PD patients to evaluate whether different modality of dialysis would influence arterial stiffness.
Moreover, we revised the description in Conclusion section as “In this study, we found that serum sclerostin greater than 208.64 pmol/L and together with old age, high values of CRP, glucose, and SBP were possible predictors to the development of AS in ESRD patients, regardless of modality of dialysis.”
Morales-Santana, S.; Garcia-Fontana, B.; Garcia-Martin, A.; Rozas-Moreno, P.; Garcia-Salcedo, J.A.; Reyes-Garcia, R.; Munoz-Torres, M. Atherosclerotic disease in type 2 diabetes is associated with an increase in sclerostin levels. Diabetes care 2013, 36, 1667-1674 Lv, W.; Guan, L.; Zhang, Y.; Yu, S.; Cao, B.; Ji, Y. Sclerostin as a new key factor in vascular calcification in chronic kidney disease stages 3 and 4. International urology and nephrology 2016, 48, 2043-2050, Qureshi, A.R.; Olauson, H.; Witasp, A.; Haarhaus, M.; Brandenburg, V.; Wernerson, A.; Lindholm, B.; Soderberg, M.; Wennberg, L.; Nordfors, L., et al. Increased circulating sclerostin levels in end-stage renal disease predict biopsy-verified vascular medial calcification and coronary artery calcification. Kidney international 2015, 88, 1356-1364 Yang, C.Y.; Chang, Z.F.; Chau, Y.P.; Chen, A.; Yang, W.C.; Yang, A.H.; Lee, O.K. Circulating Wnt/beta-catenin signalling inhibitors and uraemic vascular calcifications. Nephrology, dialysis, transplantation : official publication of the European Dialysis and Transplant Association - European Renal Association 2015, 30, 1356-1363, Jean, G.; Chazot, C.; Bresson, E.; Zaoui, E.; Cavalier, E. High Serum Sclerostin Levels Are Associated with a Better Outcome in Haemodialysis Patients. Nephron 2016, 132, 181-190, Claes, K.J.; Viaene, L.; Heye, S.; Meijers, B.; d'Haese, P.; Evenepoel, P. Sclerostin: Another vascular calcification inhibitor? The Journal of clinical endocrinology and metabolism 2013, 98, 3221-3228,
Authors should comment on role of sclerostin on CKD, and disease progression. In the discussion section, authors should discuss different origin and function of Sclerostin and DKK1 and their implications in ESRD.
Ans: Thanks for your comments. In the Discussion section, 3rd paragraph, we discussed that sclerostin accumulated as decline of renal function and restore to normal range after renal transplantation and the role of sclerostin on vascular calcification and BMD in CKD as “Bonani et al. found that the serum concentration of sclerostin rapid lowered and paralleled the improvements of renal function after receiving renal transplantation and proposed that decreased renal clearance might be responsible for the accumulation of sclerostin in late stage of CKD [30]. Additionally, there was evidence showing no correlation between vascular SOST mRNA expression and circulating sclerostin levels, which indicated less contribution of vascular-derived sclerostin [14].Through binding to LRP5/6 or forming a tertiary complex with LRP5/6, sclerostin and DKK1 respectively exerted their role in modulating the cross-talk of CKD related bone disease and vasculatures with the results of vascular smooth muscle cells trans-differentiation and vascular calcification [7,11,15,31]. Thambiah et al. and Kuo et al. conducted studies in patients of advanced renal disease and PD and found positive correlation between serum levels of sclerostin and bone mineral density (BMD) [31,32]. By intervention with an anti-resorptive agent, Miyaoka et al. found denosumab had reno-protective effects with reverse of the age related decrease of eGFR though inhibiting bone resorption and suppressing serum phosphate load in non-CKD patients [33], which indicated a possible role of manipulating sclerostin towards renal or vascular protection.”
Characteristics of control patients are not clear. On what basis Control patients were excluded from high AS, HD, and PD patients. Are control patients describe in high AS are exclusive of HD and PD patients? E.g. whether all dialysis patients were first classified as High AS patients, and all other dialysis recipient were treated as control? Please explain. A Venn diagram of different patients could be helpful in identifying different group of patients. Authors did not provide timeline of study? Also, how many days patients were examined, or if it was a single point, single day, type of study.
Ans: Thanks for your comments. We would re-describe in the Materials and Methods, Patient section as “This was a cross-sectional study which enrolled 72 PD and 122 HD patients from Hualien and Dalin Tzu Chi Hospital from March 2015 to October 2016.” And “Patients who agreed for study received anthropometric and carotid-femoral PWV examinations and had their blood samples collected, centrifuged, stored and investigated at the same day. Samples for measuring sclerostin and DKK1 were collected and stored at the same time and then examined by enzyme-linked immunosorbent assay altogether thereafter.”
Moreover, we add a Venn diagram (Figure 1) for better understanding of grouping patients into control and high AS groups.
The level of Sclerostin and DKK-1 was measured using ELISA based assay. Their values in patients should be shown as “box and whisker plots, with dots for individual value”. This value-distribution is important, as Sclerostin value in Control patients’ ranges from 89 to 175, and in High AS patients’ from 129 to 236 (Table 1). Did authors tried to group patients based on Sclerostin values (level) and determine what value of sclerostin can indicate patient condition well?
Ans: Thanks for your comments. We add a Figure shown as “box and whisker plots” to indicate the sclerostin and DKK1 levels of control and High AS groups of our patients (all patient, HD and PD patients, individually).
We did not group patients according to sclerostin levels but provided a optimal cut-off value of 208.64 pmol/L to predict the development of AS in ESRD patients.
Figure 1, legend are not self-explanatory. What does each arrowhead represent on the plot? What is the cut-off level for each parameter? What does sensitivity and 100-Specificity mean? How these plots were generated (software used) and what does they mean? Are they of diagnostic value in future? Why cfPWV was not included, as that is of prognostic value in determination of ESRD.
Ans: Thanks for your comments. We used software MedCalc® (version 12.7.2.0) to analyze ROC curve as well as SPSS for other analysis (we will add this information in the Statistical analysis section).
Each arrowhead means each patient of these ROC curve. We used ROC curve to calculate the discriminated values of variables to develop high AS. These paraments were sclerostin, age, SBP, CRP, and glucose, which showed significance of multivariate stepwise logistic regression analysis. The diagnostic ability of these paraments with discrimination threshold was shown as the “cut-off levels” in this study. The ROC curve is created by plotting the “true positive rate” (sensitivity) against the “false positive rate” (1-specificity). We showed the optimal cut-off value of each parameter along with its sensitivity and specificity. These values calculated by ROC curves could provide future diagnosis of developing high AS with positive likelihood and negative likelihood ratio.
Line 143-146, authors compared PWV with “aortic stiffness index beta”. What is the implication of it in the present study is not clear. Similarly, several other points are introduced without discussing their implication with the present study. Authors should refrain from cataloguing all the information in the discussion section; rather it should be used for discussing implications of their findings. Authors are encouraged to restructure the discussion section. The most interesting finding of the study is identification of optimal cut-off values for AS prediction. Authors should discuss this finding in more detail.
Ans: Thanks for your comments. We would re-structure the discussions with deletion of the redundant description and discussed more about the optimal cut-off values of sclerostin in this study according to your suggestions.
Manuscript needs to be proof-read well for typos, errors in grammatical usage of English, and sentence structure.
Ans: Thanks for your comments. We proof-read the whole manuscript carefully and corrected the typos, grammars and structure of sentences.

Reviewer 2 Report
Typos:
Line 20, “stiffness in in chronic” should be “stiffness in chronic”;
Line 21, “end-stage” should be unified with the same term (end stage or end-stage) in title and key words;
Line 52, “fetuin A” should be “fetuin-A”;
Line 78, “(Table 2).The” should be ““(Table 2). The”;
Line 99, Table 3’s title should not be bold;
Line 157, “DM were” should be “DM was”;
Line 177, “indoxyl slfate” Should be “indoxyl sulfate”;
Line 186, “predictive values” should be “predictive value”;
Line 246, “Body weight” should be “Bodyweight”;
Line 248, “bioimpedance were” should be “bioimpedance was”;
Line 249, “analyser” should be “analyzer”;
Line 250 and line 280, “analysed” should be “analyzed”;
Line 261, “Germany).Serum” should be “Germany). Serum”;
Line 268, “a minimum 10 minutes” should be “a minimum of 10 minutes”;
Line 275 ~ line 277, change the line spacing;
Line 285, “developing” should be “development”.
Comments:
It’s good to provide evidence that Sclerostin but not DKK-1 has the correlations with the AS in ESRD patients. It has the guidance significance for prediction and diagnostic of the AS.
Still, a multiple-center and large data pool should be collected in the followed-up studies. The conclusion now is promising to support later investigations.
Author Response
Ans: Thanks for your comments. We revised these typos according to your comments.

Reviewer 3 Report
As the topic described by Chun-Feng Wu “Serum Sclerostin but Not DKK-1 Correlated with Central Arterial Stiffness in End Stage Renal Disease Patients” is interesting and promising. There are some suggestions need to be modified carefully.
Introduction:
It would be god if prevalance of ESRD in relation to world and home country or state, that would give an idea to estimate sample size calculation
Methods:
Details of sample calculation should be explained.Anthropometric analysis:
It is advisable to include waist hip ratio which is useful indicator of abdominal obesity among females in that it reflects the risk of pulse wave velocity. Waist and hip circumferences, and waist-hip ratio were strongly correlated with higher pulse-wave velocity,independent of age, systolic blood pressure, race, and sex.
Biochemical Investigations:
The catalog no of ELISA kit should be incorporated. Sclerostin and DKK-1 measurement: Precision testing should be defined and explained to determine intra-assay and inter-assay variations.
Carotid-femoral pulse wave velocity measurements:
Time of Carotid-femoral pulse wave velocity measurements is not specified: It says only morning, while it needs to be specified and should provide information about waiting time since last food/fluid/ beverages intake or before or after breakfast, because it affects the CFP value.
I would be advisable to consider other confounding factor like alcohol intake and smoking which effect independently
Results:
In text, unit for averaged dialysis duration and CRP are missing. Table legends: Need to add all the abbreviation, statistical analysis applied, how data represented in table. Table 3 and table 4 if merge together in a single table to get clear image of analysis. As, author indicated whether HD or PD is not statistically significant (p value= 0.068; table-1), while it is significant between the groups when calculated with or without Yates' continuity correction (Need to re-analyse the data) Ambiguity in IKK-1 notation somewhere written as IKK-1 and somewhere as Dickkopf-1 In Univariate logistic regression analysis in result section, author has mentioned dialysis duration is significantly associated with AS, while p > 0.05. It is also mentioned that gender is also associated with the AS, while there is no data to justify gender correlation with AS. Result sectioned need to be explained more in details.Author Response
Introduction:
It would be god if prevalance of ESRD in relation to world and home country or state, that would give an idea to estimate sample size calculation
Ans: Thanks for your comments. We renewed the description about the prevalence of ESRD worldwide and in Taiwan as “Especially in Taiwan, there was a highest incidence and prevalence rate of end-stage renal disease (ESRD) based on the report of United States Renal Data System and the growing rates were 2.66 and 3.46 times of incidence (from 126 to 331/million populations) and prevalence (from 382 to 1322/million populations) from 1990 to 2001 according to Hwang’s report [2,3], which would further cause heavy burden of managing these comorbidities of patients.” in the Introduction section.
Foley, R.N.; Collins, A.J. The USRDS: what you need to know about what it can and can't tell us about ESRD. Clinical journal of the American Society of Nephrology : CJASN 2013, 8, 845-851 Hwang, S.J.; Lin, M.Y.; Chen, H.C.; Hwang, S.C.; Yang, W.C.; Hsu, C.C.; Chiu, H.C.; Mau, L.W. Increased risk of mortality in the elderly population with late-stage chronic kidney disease: a cohort study in Taiwan. Nephrology, dialysis, transplantation : official publication of the European Dialysis and Transplant Association - European Renal Association 2008, 23, 3192-3198
Methods:
Details of sample calculation should be explained.
Ans: Thanks for your comments. Based on references [13,14], calculation of sample sized was analyzing by MedCalc® (version 12.7.2.0). After input of α (type I error) 0.05, β (type II error) 0.9 and correlation coefficient (β) [13] or AUC [14], the needed sample size of this study was 77 [13] or 109 (number of positive or negative cases required were 45 and 64) [14], respectively. So, we enrolled all ESRD patients (120 and 74 patients of control and high AS groups) if they agree to participate this study and did not have an acute infection, malignancy, acute myocardial infarction, pulmonary edema, or heart failure at the time of blood sampling. We would add the description of sample size calculation in the Materials and Methods, Patients section.
Lv, W.; Guan, L.; Zhang, Y.; Yu, S.; Cao, B.; Ji, Y. Sclerostin as a new key factor in vascular calcification in chronic kidney disease stages 3 and 4. International urology and nephrology 2016, 48, 2043-2050 Qureshi, A.R.; Olauson, H.; Witasp, A.; Haarhaus, M.; Brandenburg, V.; Wernerson, A.; Lindholm, B.; Soderberg, M.; Wennberg, L.; Nordfors, L., et al. Increased circulating sclerostin levels in end-stage renal disease predict biopsy-verified vascular medial calcification and coronary artery calcification. Kidney international 2015, 88, 1356-1364
Anthropometric analysis:
It is advisable to include waist hip ratio which is useful indicator of abdominal obesity among females in that it reflects the risk of pulse wave velocity. Waist and hip circumferences, and waist-hip ratio were strongly correlated with higher pulse-wave velocity,independent of age, systolic blood pressure, race, and sex.
Ans: Thanks for your comments. Waist-hip ratio surely is known to be strongly correlated with PWV. However, we did not measure waist circumference (WC) of PD patients and but we re-analyzed the correlation between WC and cfPWV and central arterial stiffness of HD patients and showed as below
Linear correlation analysis: WC vs cfPWV, r = 0.154; p = 0.092
Univariate logistic regression analysis: WC vs central arterial stiffness, OR = 1.023, 95% C.I. 0.992-1.055, p = 0.146.
In HD patients, there was no significant correlation between WC and cfPWV or central arterial stiffness.
Biochemical Investigations:
The catalog no of ELISA kit should be incorporated. Sclerostin and DKK-1 measurement: Precision testing should be defined and explained to determine intra-assay and inter-assay variations.
Ans: Thanks for your comments.
The intra-assay and inter-assay coefficients of variation in the measurement for sclerostin level was 4.5% and 6.1% and DKK1 level was 2.2% and 2.9%, respectively.
Human sclerostin kit catalog no. is BI-20492
Human DKK1 ELISA kit catalog no. is BI-20413
We would revise the manuscript according to your suggestions.
Carotid-femoral pulse wave velocity measurements:
Time of Carotid-femoral pulse wave velocity measurements is not specified: It says only morning, while it needs to be specified and should provide information about waiting time since last food/fluid/ beverages intake or before or after breakfast, because it affects the CFP value.
I would be advisable to consider other confounding factor like alcohol intake and smoking which effect independently
Ans: Thanks for your comments. We confirmed the detailed of measurements and revised the description as “These measurements were performed in all patients in the supine position after a minimum of 10 minutes rest in a quiet and temperature-controlled room after breakfast in the morning”.
In this study, initially we thought of asking bad habits was not so objective and we surely did not ask patients about the habits of smoking or drinking although these these two factors were known to be associated with cfPWV. We would include these factors in future studies according to your comments.
Results:
In text, unit for averaged dialysis duration and CRP are missing.
Ans: Thanks for your comments. We revised the missing units according to your comments.
Table legends: Need to add all the abbreviation, statistical analysis applied, how data represented in table.
Ans: Thanks for your comments. We add all the abbreviations, statistical analysis used and details of table according to your comments.
Table 3 and table 4 if merge together in a single table to get clear image of analysis.
Ans: Thanks for your comments. We are wondering that you mean merge table 4 and 5 (univariate and multivariate logistic regression), so we merged these two tables for a clear image of analysis according to your comments.
As, author indicated whether HD or PD is not statistically significant (p value= 0.068; table-1), while it is significant between the groups when calculated with or without Yates' continuity correction (Need to re-analyse the data)
Ans: Thanks for your comments. We revised the data shown in Table 1 according to your suggestions.
We re-analyzed the correction between mode of dialysis and central arterial stiffness with and without continuity correction and showed that
Chi-squared test WITH continuity correction: p < 0.001 (revised Table 1)
Chi-squared test WITHOUT continuity correction: p =0.068 (original Table 1)
Ambiguity in DKK-1 notation somewhere written as DKK-1 and somewhere as Dickkopf-1
Ans: Thanks for your comments. We revised the whole manuscript with DKK-1 as the abbreviation of Dickkopf-1.
In Univariate logistic regression analysis in result section, author has mentioned dialysis duration is significantly associated with AS, while p > 0.05. It is also mentioned that gender is also associated with the AS, while there is no data to justify gender correlation with AS. Result sectioned need to be explained more in details.
Ans: Thanks for your comments.
We revised Table 4 with the missing data (gender, OR 0.563, 95% C.I. 0.314-1.011, p = 0.055).
Moreover, we used variables showed significance by univariate logistic regression analysis and those with p < 0.1 in Table 1 as possible risk factors (age, Glucose, Comorbidity, Mode of dialysis, SBP, sclerostin, CRP, gender, dialysis duration) for AS and used all these variables for multivariate stepwise logistic regression analysis. We will revise the description more precisely as “Variables showed significance in univariate logistic regression analysis and those with p < 0.1 shown in Table 1 (age, glucose, comorbidity, mode of dialysis, SBP, sclerostin, CRP, gender, and duration of dialysis) were used in multivariate logistic regression analysis.” in the Statistical analysis section.
